# The Pivotal Role of Oxytocin’s Mechanism of Thermoregulation in Prader-Willi Syndrome, Schaaf-Yang Syndrome, and Autism Spectrum Disorder

**DOI:** 10.3390/ijms25042066

**Published:** 2024-02-08

**Authors:** Claudia Camerino

**Affiliations:** 1Department of Precision and Regenerative Medicine, School of Medicine, University of Bari Aldo Moro, P.za G. Cesare 11, 70100 Bari, Italy; ccamerino@libero.it; 2Department of Physiology and Pharmacology “V. Erspamer”, Sapienza University of Rome, P.le Aldo Moro 5, 00185 Rome, Italy

**Keywords:** oxytocin, skeletal muscle, thermoregulation, Prader-Willi syndrome, Schaaf-Yang syndrome, autism spectrum disorder

## Abstract

Oxytocin (Oxt) regulates thermogenesis, and altered thermoregulation results in Prader-Willi syndrome (PWS), Schaaf-Yang syndrome (SYS), and Autism spectrum disorder (ASD). PWS is a genetic disorder caused by the deletion of the paternal allele of 15q11-q13, the maternal uniparental disomy of chromosome 15, or defects in the imprinting center of chromosome 15. PWS is characterized by hyperphagia, obesity, low skeletal muscle tone, and autism spectrum disorder (ASD). Oxt also increases muscle tonicity and decreases proteolysis while PWS infants are hypotonic and require assisted feeding in early infancy. This evidence inspired us to merge the results of almost 20 years of studies and formulate a new hypothesis according to which the disruption of Oxt’s mechanism of thermoregulation manifests in PWS, SYS, and ASD through thermosensory abnormalities and skeletal muscle tone. This review will integrate the current literature with new updates on PWS, SYS, and ASD and the recent discoveries on Oxt’s regulation of thermogenesis to advance the knowledge on these diseases.

## 1. Introduction

Andrea Prader, Heinrich Willi, and Alexis Labhart were the first to describe Prader-Willi syndrome in 1956 in a short article where they highlight the main features of this syndrome as obesity and excessive food consumption. This happened one year after the award of the Nobel Prize for the discovery of oxytocin (Oxt) to Vincent DuVignery in 1955. However, knowledge has vastly expanded since then, and this syndrome is now known to be a rare and severe neurodevelopmental disorder characterized by cognitive disabilities, behavioral problems, and a specific hypothalamic dysfunction causing hyperphagia and endocrine abnormalities. Further studies helped the understanding of mortality in individuals with PWS and the development of therapies to improve overall quality of life [1]. The syndrome results from the loss of expression of paternal genes in the PWS region of chromosome 15 at positions q11-13 [2]. However, recent data indicate the involvement of the hormone neurotransmitter oxytocin (Oxt) in the etiology of PWS [3]. Oxt is a secreted by the paraventricular (PVN) and supraoptic (SON) nuclei of the hypothalamus and is involved in several physiological functions such as energy regulation and appetite [4,5]. Indeed, an abnormality in the Oxt system is often reported, especially in PWS children [6]. PWS infants are hypotonic, and the obesity characteristic of PWS is also related to reduced energy expenditure because of hypotonia. Oxt regulates energy metabolism [7]. About 15 years ago, based on our lab results, we suggested that mice homozygous for deletions of Oxt or its receptor (Oxtr) show late-onset obesity with no changes in food consumption [8,9]. Oxt/Oxtr^−/−^ mice showed normal body weight at birth and late-onset obesity around reproductive age [3]. Oxtr^−/−^ mice were hypothermic compared to control mice, and this was the first evidence that Oxt regulates thermogenesis [10,11,12]. Indeed, a thermogenic challenge increases the expression of Oxtr in the brain and in the slow-twitch muscle as the circulation of Oxt decreases following a negative feedback in the brain [13]. According to the results of these studies, we hypothesized that the late-onset obesity of Oxt/Oxtr^−/−^ mice was caused by muscle depotentiation and fat infiltration rather than increased food consumption. Conversely, Oxt infusion increases body temperature [14]. Specifically, in our model of induced thermogenesis, the expression of Oxtr increases in the hypothalamus as circulating Oxt decreases in mice. Conversely, in PWS individuals, a decrease in Oxtr expression as circulating Oxt increases has been reported in a post-mortem analysis of the brain. The phenotype of the cold-stressed mice in our model is a mirror image of PWS individuals. Recent studies indicate autism spectrum disorder (ASD) individuals present sensory deficits [15,16,17,18]. PWS is also characterized by thermosensory abnormalities [19,20] as well as Schaaf-Yang syndrome (SYS), which has several overlapping features with PWS but is caused by pathogenic/truncating variants of the melanoma antigen L2 gene (MAGEL2), and a diagnosis of ASD is more frequent [21,22]. Several articles have reviewed the evidence supporting therapy options for the treatment of hyperphagia and obesity in PWS [1,23] as well as PWS and anti-obesity medication [1,23]. These articles were written by performing a literature search using PubMed, the word PWS, and drugs including Oxt in the year 2021. However, the new knowledge that Oxt regulates muscle mass, decreases proteolysis in skeletal muscle [24,25], and triggers muscle contraction by thermoregulation [13,26] was established very recently, after 2021. Based on this rationale, our aim to further advance the knowledge on PWS drove us to integrate the current knowledge with this review, in which we explain the latest data on Oxt’s regulation of thermogenesis and skeletal muscle metabolism and review the sensory deficits typical of PWS, SYS, and ASD to provide a translational meaning to our studies.

## 2. Prader-Willi Syndrome and Schaaf-Yang Syndrome Features and Similarities

PWS is a genetic disorder resulting from the lack of expression of imprinted genes in the paternally derived region of the chromosome: 15q11-q13 [1]. The main characteristic of PWS is obesity and hyperphagia. PWS individuals undergo five nutritional phases during life that can be classified as phase 0, in which intrauterine growth is restricted and very few movements are detected by the mother, and four subsequent phases characterized by hypotonia and obesity, as previously described [2]. The genetic characteristics and the five nutritional phases of PWS are extensively described in Table 1.

As shown in Figure 1A in healthy individuals, the Oxt system is functional. The extreme low temperature increased Oxtr in PVN and the Soleus muscle and decreased circulating Oxt. This mechanism triggers the “oxytonic contractions” that potentiate and augment the tonicity of the slow-twitch muscle. As shown in Figure 1B, the Oxt system is dysfunctional in Prader–Willi syndrome individuals, where a reduced expression of Oxtr in PVN leads to increased Oxt secretion by the posterior pituitary due to the loss of negative feedback. In the healthy individual, the red arrow represents the negative feedback that decreases circulating Oxt following the increase of Oxtr in PVN. The blue arrow represents the positive feedback that increases Oxtr in PVN following the decrease of circulating Oxt. In the Prader-Willi individual, the red arrow represents the negative feedback that decreases Oxtr in PVN after the increase of circulating Oxt. The blue arrow represents the positive feedback that increases circulating Oxt after the decrease of Oxtr in PVN [13]. Figure 1C shows the effect of Oxt intranasal administration on infants with Prader-Willi syndrome. Oxt administration in infants causes a general improvement in muscle tone and weight because, at a very early age, Prader-Willi syndrome individuals are hypotonic, but obesity is not yet established [27,28,29]. As shown in Figure 1D, in Schaaf-Yang syndrome individuals, hyperphagy and obesity are less frequent, and no oxytocin measurement is available. However, autism spectrum disorder is present more frequently in Schaaf-Yang syndrome individuals, compared to those with Prader-Willi syndrome. As shown in Figure 1E, individuals with autism spectrum disorder have sensory abnormalities that manifest with hyper- and hypo-reactivity to smell, taste, audition, vision, and tactile sensitivity, recently included in the DSM-V. In autism spectrum disorder, Oxt appears dysregulated. Differently from Prader-Willi syndrome individuals, these abnormalities in autism spectrum disorder are not caused by hypothalamic syndrome.

PWS individuals have a shorter life expectancy caused by several morbid conditions, including febrile illness [27]. These is no cure for PWS, and weight control is one of the main targets of treatment. Growth hormone (GH) replacement therapy, which is the only approved treatment in USA, can improve body composition, physical strength, and cognitive level, and the beneficial effects persist after the treatment is discontinued [28,29]. In parallel with the development of nutritional phases, behavioral issues arise with cognitive impairment and emotional deregulation in PWS. These clinical features are caused by the abnormal development and function of the hypothalamus. Other characteristics of PWS include severe hypotonia in infants, which starts improving in early childhood, although it remains later in life [32,33,34]. The SNORD116 gene appears to drive the phenotype of PWS [35]. Moreover, individuals carrying a mutation of MAGEL2 have been identified in a cohort of patients with ASD [36], and a new syndrome has subsequently been classified as Schaaf-Yang syndrome [22]. In these patients, ASD is more frequent than in PWS, but obesity is less frequent [36]. However, in PWS individuals, diabetes is rare, even in the presence of obesity [37]. Moreover, patients with uniparental disomy show a different metabolic profile compared with patients with a deletion. Adult PWS individuals show a decreased expression of Oxt-secreting neurons both in number and volume in the PVN compared to healthy controls, as shown by histomorphometry, while plasma Oxt concentrations appear to be increased in the cerebrospinal fluid of PWS individuals [38,39,40]. Abnormalities in Oxt expression have also been reported in an animal model of PWS as MAGEL2 and necdin knockout mice [41]. Finally, clinical trials on infants treated with Oxt show very promising results [42]. Treatment with Oxt later in life has less consistent effects, probably because of receptor desensitization and overdosing or because the obesity is already established, as previously described [3,43]. Ghrelin, also called the hunger hormone, is produced by the gastric mucosa and is present in an acylated form with hunger effects and in an unacylated form with opposite effects [44]. PWS individuals show hyperghrelinemia and a low concentration of brain-derived neurotrophic factor, which is also related to obesity [45,46,47]. PWS individuals have severe GH deficiency, and this is higher in patients with uniparental disomy than in patients with deletion [48]. In conclusion, impaired Oxt and Ghrelin systems described in this syndrome, together with GH dysfunction, are the main hallmark of this disease [2]. Schaaf-Yang syndrome (SYS) was originally considered Prader-Willi-like syndrome because it is caused by a nonsense frame shift mutation in the maternally imprinted and paternally expressed MAGEL2 gene located in the Prader-Willi critical region 15q11–15q13, also called the MAGEL2 loss of function [22,49]. SYS manifests as developmental delay/intellectual disability, hypotonia, feeding difficulty, and ASD [50]. PWS and SYS overlap for several but not all clinical features. SYS patients exhibit a combination of short stature, high fat mass, and low IGF-1 levels, suggesting a GH deficiency like PWS. Indeed, SYS and PWS individuals present some endocrinological overlap, as low IGF1, also in the presence of normal weight and GH deficiency, as seen in PWS patients [51]. Fasting ghrelin is elevated in both syndromes, and patients present abnormal BMD and scoliosis. Ghrelin is also high in SYS patients with normal weight, and diabetes mellitus is more common than in PWS individuals. In conclusion, the combination of short stature, elevated fat mass/reduced lean mass, low bone density, and low IGF1 levels suggest a GH deficiency similar to that seen in PWS individuals [49]. However, while there are many shared characteristics, some clinical features are typical of SYS, like interphalangeal joint contractures. While PWS individuals present a BMI indicating overweight and obesity, SYS individuals present a lower BMI incongruous with the high fat mass of this patients, and while PWS individuals develop hyperphagia and obesity, only a minority of SYS individuals do so. These differences are probably caused by muscular hypotonia, albeit studies are lacking. Lastly, SYS individuals show a higher incidence of ASD compared to PWS individuals [36]. Oxt expression in SYS has yet not been established; however, Oxt has been administrated in SYS patients to treat ASD, which is frequent with this syndrome [22]. These differences are explained in Table 1.

Management of PWS and SYS is mostly symptomatic. The mechanistic explanation of these diseases remains elusive. PWS and SYS are hypothalamic diseases, and, in this context, MAGEL2 evolved as a hypothalamic regulator of regulated secretion [52]. Multiple reports have demonstrated Oxt abnormalities in PWS patients including fewer Oxt-producing neurons and altered Oxt levels in plasma [38,40]. However, dysregulation of the Oxt system may go beyond the altered expression of Oxt in these syndromes and may affect muscle functions and thermoregulation, as explained below.

## 3. Thermosensory Abnormalities in Prader-Willi Syndrome and Autism Spectrum Disorder: Has the Holy Grail Been Overlooked?

Temperature regulation is impaired in PWS individuals since they show significant changes in thermoregulation compared to siblings or controls [53]. Oxt/Oxtr^−/−^ mice have impaired thermoregulation, resulting in thermosensory deficits and the incapacity to maintain a stable temperature. Although the scope of this article is not to focus on animal models, it is worth noting that PWS arises from the loss of expression of a critical genetic region on chromosome 15 (q11.2-13), a region that is largely conserved on chromosome 7 in mice [54]. The PWS locus is large and includes a number of known protein-coding and non-coding genes including SNORD116. The lack of a SNORD116 gene cluster is present in PWS individuals with microdeletions within this locus. Snord116 knockout mice have been generated, and the metabolic phenotype of germline Snord116^−/−^ mice is influenced by thermoregulation. Indeed, while Snord116^−/−^ mice housed at 22 °C exhibited a PWS-like phenotype as hyperphagia and changes in energy expenditure, most of these features, with the exception of low body weight, were rescued when mice were housed at thermoneutrality of 30 °C [55]. The role of Snord116 in the development of the PWS-like phenotype depends also on dysfunctions of the NPY system, likely caused by disrupted regulation of the orexigenic NPY/agouti-related peptide (AgRP) neurons and the anorexigenic POMC/cocaine and amphetamine regulated transcript (CART) neurons, which are located in the Arc. The deregulation of NPY mRNA in the Snord116^−/−^ mice is no longer observed when mice are held under thermoneutral conditions. This suggests that both the expression of NPY and the function of Snord116 are dependent on thermoregulation [54]. Moreover, reintroduction of the SNORD116 gene in Snord116^−/−^ mice in the mid-hypothalamous and at an early age led to upregulation in energy expenditure but only at thermoneutral conditions [54]. Single gene mutations constitute a rare subset of PWS individuals and display an incomplete phenotypic spectrum. For this reason, PWS-IC^del^, a mouse model for full PWS, was generated [56]. IC deletion in this mouse model results in the compete loss of paternal gene expression from the PWS interval, including its expression in the central nervous system. Interestingly, the metabolic phenotype of PWS is replicated in adult PWS-IC^del^ mice, including hyperghrelinemia and hyperphagia. However, these mice show intra-abdominal leanness at 22 °C. Maintenance of thermoneutral conditions at 30 °C suppressed the thermogenic activity in this mice, although reduced excessive food consumption had no effects on increasing abdominal fat mass [56]. The hypothesis that hyperphagia may be caused by impairment in the anorexic Oxt circuitry arose at the time of these studies, although the quantification of Oxt expression in these mice has not been established. In conclusion, it seems that a gene linked to thermoregulation influences the manifestation of the PWS phenotype. A small but statistically significant proportion of PWS children have been reported to experience persistent or episodic hypothermia, whereas fingertip temperature is unaltered [53,57,58]. The altered perceptions in PWS are related to a reduction in sensory neurons in the dorsal root ganglia due to the absence of necdin expression rather than obesity [59]. About 90% of ASD individuals have sensory abnormality that manifests with hyper- and hypo-reactivity to smell, taste, audition, vision, and especially tactile sensitivity. Recently, this sensory processing deficit was included for the first time among the international diagnostic criteria of autism in a revision of the Diagnostic and Statistical Manual of Mental Disorders (DSM-V). The sensory deficit can influence the perception of the external word and consequently social behaviour [60]. Oxt is released also in response to tactile stimuli, and Oxt abnormalities have been found in the plasma level of autistic children [61]. Abnormalities in the gene that encodes for Oxtr [62] as well as in Oxt has been found in ASD [63]. This is why we hypothesize that Oxt regulation of thermogenesis can cause sensory abnormalities in ASD. Moreover, children with ASD present altered thermal thresholds with reduced sensitivity to warmth, coolness, and heat, but not pain threshold. Paradoxical heat sensation with a sensation of burning heat in response to cold stimulation has also been reported, indicating disruption of thermosensory integration [16]. This evidence is of special note since ASD individuals, differently from PWS individuals, do not have hypothalamic syndrome that can per se cause temperature fluctuations and sensory abnormalities. However, a more comprehensive study of ASD is not the scope of this work. Oxt’s regulation of thermogenesis may be responsible for the sensorial functionality and body temperature regulation through muscle contraction in healthy individuals, while a dysfunctional Oxt system can cause sensory abnormalities and muscle hypotonicity as seen in PWS, SYS, and ASD.

## 4. New Evidence Establishing Oxytocin in the Maintenance of Muscle Metabolism and Tonicity

Oxt is related to the regulation of energy and metabolism [8,9], as Oxt concentration increases in pregnancy and lactation in females, and Oxt triggers aggressive behavior and augments muscle strength. Evolutionarily, this is of paramount importance given that labor is needed for the protection of offspring when the offspring is most vulnerable to predators and Oxt concentration in plasma is at its peak [64,65]. Consistent with these studies, we hypothesized that Oxt may increase muscle tone to ensure a better response to the “fight response” [66]. This initial evidence was further confirmed in the coming years. Indeed, although it has been previously demonstrated that Oxtr stimulation can control skeletal muscle mass in vivo, the intracellular mechanism that mediated this effect is still poorly understood. The activation of Oxtr directly induces skeletal muscle protein sparing effects through a Gαq7/IP3R/Ca^2+^-dependent pathway and crosstalk with AKT/FoxO1 signaling, which decreases the expression of genes related to atrophy, as in the case of LC3, and muscle proteolysis [25]. The loss of muscle mass is related to several pathological conditions and is dominant also in PWS individuals. The loss of muscle mass and atrophy is often caused by proteolysis [67], although it can be treated with hormones and other pharmacological intervention [68]. Oxt also has a positive tonic effect on oxidative muscle and on the myocardium, and serum Oxt is positively related to lean mass [69,70,71,72,73,74]. Moreover, Oxtr has an inhibitory effect on muscle proteolysis dependent on Ca^2+^-AKT signaling. The mechanism by which Oxt exerts its anti-catabolic actions on protein metabolism in oxidative skeletal muscle is mediated by Ca^2+^ efflux through IP3 and the coupling between Oxtr and Gαq protein [25,75]. These data support the anti-diabetogenic action of this neuropeptide, which can be explained also by the maintenance of skeletal muscle mass. To add further credence to the Oxt effect on muscle mass, a recent study described the association between serum Oxt concentration and lean and fat mass after metabolic and bariatric surgery and sleeve gastrectomy [24]. Metabolic and bariatric surgery is an in vivo model that can be used to study Oxt concentration related to fat and lean mass [24]. In this observational study, baseline serum Oxt concentration was positively associated with changes in weight and BMI over 12 months, and Oxt concentration post-surgery was positively associated with changes in lean mass but not fat mass. This reduction, however, was no longer significant after normalizing for changes in BMI, supporting the hypothesis that Oxt levels reflect energy availability and appetite with higher levels in individuals with high BMI versus normal weight individuals as in PWS individuals. Higher baseline Oxt levels are associated with a greater degree of weight loss after sleeve gastrectomy, and Oxt concentration can predict the percentage of weight loss after surgery. Of note is that Oxt administration can help rebuild muscle lean mass after sleeve gastrectomy, and these studies can be applied to diseases characterized by obesity and hypotonia, like PWS.

## 5. Oxytocin’s Regulation of Thermogenesis and the Cold Stress Model

Oxt is anabolic to muscle, and Oxt anti-obesogenic effects are related to its positive effects on muscle mass [76,77,78]. In this regard, it is possible that there is a feedback loop between the hypothalamous and muscle that regulates the rate of central Oxt release and muscular Oxtr expression in response to situations requiring increased musculoskeletal performance [3,13]. In our studies, we characterized Oxt and Oxtr expression in different tissues after a cold stress (CS) challenge in mice by an ex vivo and in vitro approach. We showed that Oxtr expression increases in Soleus (Sol) muscle and in the brain after the CS challenge while circulating Oxt decreases in mice [13], and this phenotype can be linked to PWS, as we explain in the following section. The cold stress model is explained in Table 2 and has been previously shown [13]. 

## 6. The Cold Stress Model and the Link with Prader–Willi Syndrome, Schaaf-Yang Syndrome, and Autism Spectrum Disorder

Oxt is a master gene in the regulation of thermogenesis [13]. This is why we postulate that the disruptions of the Oxt system manifests in PWS, SYS, and ASD in the following manner. Regarding PWS, post mortem analysis indicates that PWS individuals show less Oxtr in the brain [78] and higher circulating Oxt compared to healthy controls [41]. This phenotype of PWS individuals showing less Oxtr in the brain and high circulating Oxt is a mirror image of the cold-stressed mice in our model of a thermogenic challenge. Indeed, cold-stressed mice present a higher expression of Oxtr in the brain, specifically in the PVN and hippocampus and in Sol and lower circulating Oxt compared to non-cold-stressed mice, indicating a higher tonicity of the slow-twitch muscle [13]. The mirror image of the phenotype of cold-stressed mice shows low Oxtr in the brain, high circulating Oxt, and hypotonic muscles, as in PWS individuals (Figure 1). This phenotype can be rescued by the intranasal administration of Oxt, but only when children are treated at a very early age [27]. The negative feedback between the brain and the peripheral organs that trigger a higher expression of Oxtr in the brain and of Sol and low circulating Oxt was called in our laboratory “the Oxytonic effect” [13]. It is worth postulating that, rather than treating PWS individuals with Oxt, it may be useful to treat them with an Oxt antagonist to break down the positive feedback between the brain and the periphery to reset the oxytonic effect. Regarding ASD, the abnormal expression of the Oxt gene that acts as a master gene in the regulation of thermogenesis may trigger the sensory abnormalities seen in ASD [15,16,17]. Indeed, very recent data indicate that ASD individuals experience sensory abnormalities in vision, audition, tactile, and postural functions, and these features have been included in the DMS-V as diagnostic criteria of ASD. Specifically, ASD presents paradoxical heat sensation where a cold stimulus is perceived as burning hot [15,16,17]. Such a distorted perception of the external world can bias the perception of the internal state and consequently affects the interaction of ASD individuals with the outside world. It is worth noting that, in ASD, obesity and hyperphagia are not present, which means that the sensory abnormalities are not caused by the hypothalamic syndrome present in PWS individuals. However, ASD individuals show Oxt abnormalities, like in PWS individuals, and benefit from Oxt administration [61,62,63]. Indeed, hypothalamic syndrome can itself trigger the sensory deficits, abnormal thermoregulation, and episodes of high fever or low body temperature [20,57,79]. Moreover, PWS individuals have low muscle tone, while cold-stressed mice have high Oxtr expression in Sol [13,80]. This means that PWS infants are hypotonic and then progress with sarcopenic obesity in childhood and adulthood. Regarding SYS individuals, they are hypotonic at birth but hyperphagia and obesity are less frequent, differently from PWS. SYS individuals have a higher incidence of ASD compared to PWS. This article focuses more on PWS than on SYS because no measurement of Oxt in SYS individuals is available to date. The *MAGEL2*-KO mouse is a model of SYS, in which Oxtr is downregulated at an early age, consistent with previous studies [13], but Oxtr is upregulated later in life [30]. The administration of Oxt has been shown to restore normal Oxtr levels [30]. The dysfunctional Oxt system in PWS and SYS individuals may go beyond the Oxt/Oxtr expression level and affect the overall Oxt neuron activity by the imbalance of synaptic excitation/inhibition, as seen in Oxt-neurons in which *MAGEL2* has been inactivated [81]. SYS individuals are hypotonic, like PWS individuals, and present a higher incidence of ASD, meaning that the Oxt dysfunction may occur at different degrees in the three syndromes considered in this article. Indeed, hypothalamic grey matter volume (GMV) shows no group-related differences between individuals with or without ASD; however, GMV shows a positive association with Oxt in ASD individuals and a negative association in healthy individuals [31]. This underlines the presence of feedback between the brain and circulating Oxt, consistent with our cold-stress model. It is important to underline that thermogenesis may increase the volume of certain brain areas [31]. Oxt regulates thermogenesis [13], and when Oxt is lacking or dysfunctional, such an increase in brain volume does not take place, leading to sensory deficits. This may be the case of ASD individuals that suffer sensory deficits, but these deficits are not caused by hypothalamic syndrome, as they are in PWS and SYS cases. Indeed, the sensory deficits and abnormalities observed in ASD, PWS, and SYS can be caused by an insufficient increase in the volume of the appropriate brain area and the expression of Oxtr. We want to underline that, taken together, this evidence suggests a connection between PWS, SYS, ASD, and Oxt regulation of thermogenesis, but further studies are required.

## 7. Conclusions

In summary, the question of whether PWS, SYS, and ASD are the manifestation of dysfunctions in Oxt regulation of thermogenesis remains open. The evidence that Oxt is a master gene that regulates thermogenesis may be related to the sensory deficits seen in ASD individuals. An improved understanding of Oxt abnormality in PWS, SYS, and ASD is needed, and previous knowledge regarding these diseases must be integrated with the recent discoveries regarding the impact of Oxt regulation on thermogenesis, muscle contraction, and sensory abnormalities. Oxt’s regulation of thermogenesis, Oxt-driven muscle contraction, and episodes of hypothermia or hyperpyrexia are well related and crucial in the manifestation of these pathologies. Our hope is that these studies will be applied to clinical research and improve the quality of life of patients affected by these syndromes. 

## Figures and Tables

**Figure 1 ijms-25-02066-f001:**
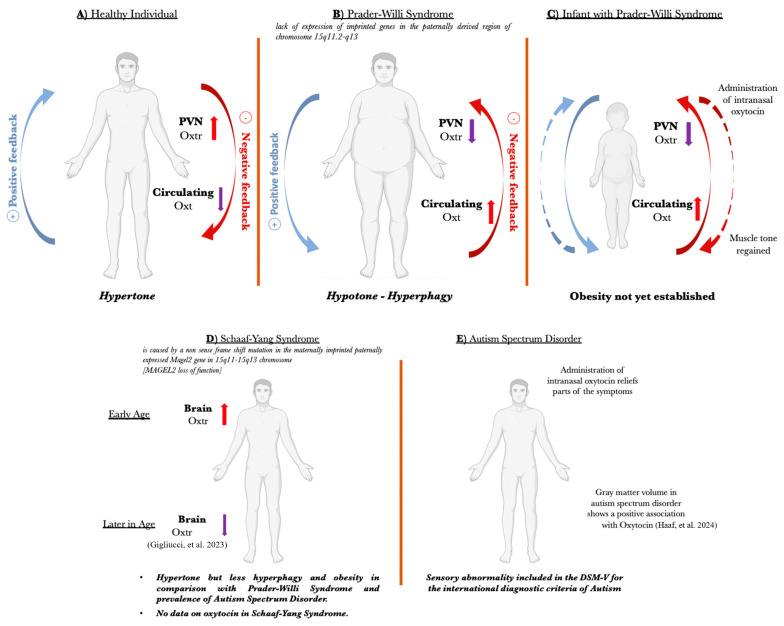
Schematic representation of Oxytocin’s regulation of thermogenesis. (**A**) The physiological mechanism in healthy individuals and the dysfunctional mechanism in (**B**) a Prader-Willi syndrome individual, (**C**) an infant with Prader-Willi syndrome treated in early life with intranasal Oxytocin [29], (**D**) an individual with Schaaf-Yang syndrome [30], and (**E**) an individual with autism spectrum disorder [31].

**Table 1 ijms-25-02066-t001:** Genetics and clinical features of Prader-Willi syndrome and Schaaf-Yang syndrome.

**Prader-Willi Syndrome**	**Genetic**	The syndrome results from the loss of expression of paternal genes in the Prader-Willi syndrome region of chromosome 15 at positions q11-q13. These genes are maternally imprinted, which means they are only expressed from the paternal chromosome. Genetic sub-types: (1) paternal deletion (type I or type II, or athypical deletions of different lengths depending on the proximal chromosomal breackpoint), (2) maternal uniparental disomy, (3) rare imprinting defects, and (4) various and very rare chromosomal rearrangements such as translocations involving the Prader-Willi syndrome region.
	**Metabolic**	Phase 0: Intrauterine grow is restricted. Phase 1: At birth, the infant is hypotonic. Sub-phase 1a: (aged 0 to 9 months) characterized by difficulty feeding and failure to thrive Sub-phase 1b: (aged 6 months to 2 years) The infants grow steadily along a growth curve, and weight increases at a normal rate. Phase 2a: (aged 2–4 years) associated with excessive weight gain without changes in appetite or caloric intake Phase 2b: (aged 4–6 years) increased interest in food Phase 3: hyperphagia, food-seeking, and insatiable appetite Phase 4: adults; no insatiable appetite; individuals feel full.
	**Other** **characteristics**	Unexplained hyperpirexia/hypothermia, short stature/grow hormone deficiency, hypothyroidism, impaired glucose tolerance/diabetes mellitus, high level of total cholesterol and low-density lipoprotein (LDL) cholesterol. Scoliosis. Low oxytocin receptor density was reported in the brain of Prader-Willi individuals, while circulating oxytocin has been reported higher than in controls.
	**Behavioural phenotype**	Deficits in social skills, learning abilities, autistic spectrum disorder.
**Schaaf-Yang Syndrome**	**Genetic**	Nonsense and frameshift pathogenic variants in the maternally imprinted and paternally expressed MAGEL2 gene. MAGEL2 is an intronless gene in the Prader-Willi syndrome domain on chromosome 15q11-15q13 that encodes a protein important for endosomal protein trafficking.
	**Clinical** **features**	Phenotypic overlap between individuals with MAGEL2 loss of function and those with Prader-Willi syndrome includes neonatal hypotonia, feeding difficulties, weight gain, developmental delay/intellectual disability, and hypogonadism. Clinical features distinct from Prader-Willi syndrome includes interphalangeal joint contractures (82% of patients), the prevalence of autism spectrum disorder (27% in Prader-Willi syndrome vs. 77% of Schaaf-Yang syndrome), only a minority of Schaaf-Yang syndrome patients develop hyperphagia and morbid obesity.
**Prader-Willi Syndrome** **vs.** **Schaaf-Yang Syndrome**	**Ghrelin**. Patients with Schaaf-Yang syndrome show an even further elevated average overnight fasting ghrelin level with a consistently high level compared with controls and with Prader-Willi syndrome individual references, although SYS patients rarely develop hyperphagia.	
	**Glucose Tollerance and diabetes mellitus**. Schaaf-Yang syndrome patients present an increased prevalence of glucose intolerance and diabetes mellitus compared with Prader-Willi syndrome.	
	**IGF-1**. Patients with Schaaf-Yang syndrome express depressed levels of IGF-1 similar to Prader-Willi syndrome.	
	**Thyroid Hormones**. Patients with Prader-Willi syndrome individuals are more likely to develop hypothyroidism than controls, while Schaaf-Yang syndrome individuals do not display evidence of central hypothyroidism.	
	**Oxytocin**. No data on oxytocin in Schaaf-Yang syndrome are available to date (Duan clinical case medicine 2022).	
	**Scoliosis**. Scoliosis is often common in Prader-Willi syndrome patients with a prevalence of 40–80% and varies in age onset and severity. MAGEL-2 could be the scoliosis-determining gene in Prader-Willi syndrome. Schaaf-Yang syndrome patients present a slightly lower frequency of scoliosis with a prevalence around 33%; however, scoliosis is progressive and can onset later in life.	
	**Bone mineral density and body composition**. Individuals with Prader-Willi syndrome display low bone mineral density (BMD) that can result in a high risk of fractures and osteoporosis. Body mass index (BMI) shows a high prevalence of overweight/obesity in Prader-Willi syndrome, while BMI in Schaaf-Yang syndrome individuals is lower.	

**Table 2 ijms-25-02066-t002:** Oxytocin and oxytocin receptor expression in different tissues after exposure to 6 h or 5 days of cold stress in wild-type mice.

(A) **In vitro studies**					
	**Oxt 6 h**	**Oxtr 6 h**		**Oxt 5 d**	Oxtr 5 d
*Soleus*	***	***		***	*+*
*Tibialis Anterioris*	***	***		***	*+*
*Brown Adipose Tissue*	*-*	***		***	***
*Bone*	*+*	***		***	***
*Brain*	***	*+*		***	*+*
(B) **In vitro studies: Cold stress in Myosin**					
- *Soleus*					
Myhc2b			−		
Myhc1			***		
- *Tibialis Anterioris*					
Myhc2a			−		
Myhc1			***		
(C) **Ex vivo studies**					
	**Oxt 6 h**	**Oxtr 6 h**		**Oxt 5 d**	**Oxtr 5 d**
*Hypothalamus*	***	***		***	***
*Supraoptical Nucleus*	***	*+*		***	*+*
*Paraventricular Nucleus*	***	*+*		***	*+*
*Hippocampus*	***	***		***	*−*
*Circulating Oxytocin*	decreases				

* no changes detected; “−” indicates a significative decrease comparing to untreated control; “+” indicate a significative increase comparing to untreated control. (A) Oxtr regulates the coordinated gene response to cold stress through a feed-forward loop in brain as shown by a regression study in which the elimination of Oxtr expression gene data in brain causes the loss of gene expression at thermoneutrality but not after cold stress [12]. (B) The ratios Myhc1/Myhc2b and Myhc1/Myhc2a in Sol and Tibialis anterioris (TA) muscles are increased after cold stress suggesting that thermogenic challenges affect the slow-twitch muscle and shifts TA muscle toward the slow-twitch phenotype while potentiating the slow-twitch phenotype of Sol. Low circulating Oxt levels are required after long-term cold stress. However, the up-regulation of Oxtr in brain and Sol balances the decreased circulating Oxt (Oxytonic effect) [13]. (C) Cold stress increases Oxtr in Paraventricular nucleus of hypothalamus as shown by histomorphometry and decreases circulating Oxt in a feed-back/feed-forward loop with brain [13].

## Data Availability

Not applicable.

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
