# Peer review of "The Pivotal Role of Oxytocin’s Mechanism of Thermoregulation in Prader-Willi Syndrome, Schaaf-Yang Syndrome, and Autism Spectrum Disorder"

_ijms, 2024, doi:10.3390/ijms25042066_

Round 1

Reviewer 1 Report

Comments and Suggestions for Authors

The article titled "The Central Role of Oxytocin's Thermoregulatory Mechanism in Prader-Willi Syndrome, Schaaf-Yang Syndrome, and Autism Spectrum Disorder" serves as a comprehensive synthesis of recent research findings while concurrently proposing a novel hypothesis, thereby establishing a foundational framework for the treatment of these conditions and the expansion of our understanding regarding these pathological states.

The authors have indeed demonstrated a commendable degree of diligence in their examination of the subject matter. However, it is imperative to address the issue of clarity surrounding the newly posited hypothesis. It is suggested that the inclusion of a visual representation, such as a diagram, elucidating the perturbations within the Oxytocin thermoregulatory mechanism in the context of Prader-Willi Syndrome (PWS), Schaaf-Yang Syndrome (SYS), and Autism Spectrum Disorder (ASD) through the lens of thermosensory disturbances and skeletal muscle tone would significantly enhance the comprehensibility of the proposed concept.

Furthermore, in order to augment the accessibility and readability of the manuscript, it is advisable to consider incorporating additional figures or tables. These visual aids can serve as invaluable tools in conveying complex information succinctly and promoting a more lucid exposition of the research findings, ultimately facilitating a deeper understanding of the subject matter.

Author Response

                                                                                           December 28 2023

Reviewer # 1

Dear respected reviewer number 1, thank you for the positive comments to our manuscripts.

The article titled "The Central Role of Oxytocin's Thermoregulatory Mechanism in Prader-Willi Syndrome, Schaaf-Yang Syndrome, and Autism Spectrum Disorder" serves as a comprehensive synthesis of recent research findings while concurrently proposing a novel hypothesis, thereby establishing a foundational framework for the treatment of these conditions and the expansion of our understanding regarding these pathological states.

The authors have indeed demonstrated a commendable degree of diligence in their examination of the subject matter. However, it is imperative to address the issue of clarity surrounding the newly posited hypothesis. It is suggested that the inclusion of a visual representation, such as a diagram, elucidating the perturbations within the Oxytocin thermoregulatory mechanism in the context of Prader-Willi Syndrome (PWS), Schaaf-Yang Syndrome (SYS), and Autism Spectrum Disorder (ASD) through the lens of thermosensory disturbances and skeletal muscle tone would significantly enhance the comprehensibility of the proposed concept.

Dear respected reviewer number 1, figure1 has been integrated with 2 new panels regarding Schaaf-Yang syndrome and autism spectrum disorder and the legend of this figure has been expanded.

Furthermore, in order to augment the accessibility and readability of the manuscript, it is advisable to consider incorporating additional figures or tables. These visual aids can serve as invaluable tools in conveying complex information succinctly and promoting a more lucid exposition of the research findings, ultimately facilitating a deeper understanding of the subject matter.

Dear respected reviewer number 1, a new Table 2 has been created and incorporated in the articles with a new legend.

Reviewer 2 Report

Comments and Suggestions for Authors

This review article on oxytocin is part of a special issue on Oxytocin in IJMS. Dr. Camerino has nine other publications on oxytocin, the oxytocin knockout mice, and the oxytocin receptor, with four of these as review papers.  The paper is poorly written and  does not fully provide an overview of what is known about Oxt in Prader-Willi syndrome (PWS), Schaaf-Yang Syndrome (SYS) and Autism Spectrum Disorder (ASD), instead focusing more on PWS, and having significant overlap with another manuscript published by the same author in 2023. Suggestions and comments below are provided.

1.     I would like to point out that there may be significant overlap with Dr. Camerino’s article and another one she published in 2023.  In reviewing Dr. Camerino’s previous work in the area, I came across a very similar article, PMCID: PMC10297258. Even within the abstract, the line in the abstract “Prader–Willi Syndrome (PWS) is a genetic neurodevelopmental disorder that is caused by either the 13 deletion of the paternal allele of 15q11-q13, maternal uniparental disomy of chromosome 15 or defects in the chromosome 15 imprinting centre and is characterized by hyperphagia with significant 15 risk of obesity, low skeletal muscle tone and a variety of maladaptive behaviours and autistic spectrum disorder (ASD) for this reviewed manuscript is nearly identical to a line in her published article “…is characterized by cognitive impairment, hyperphagia and low metabolic rate with significant risk of obesity, as well as a variety of other maladaptive behaviours and autistic spectrum disorder (ASD).

a.     In addition, Figure 1 from both articles are very similar, with just the addition of the infantile phase of PWS included in this article being reviewed. The text in Figure 1 in this article does not appear to be formatted correctly with extra spacing in the words within the figure (i.e. O xt and C irculating)

b.     Using Figure 1 in both articles, the author goes on to describe nearly the same information for PWS and oxt.

c.     Since this is an article on all three syndromes, the figure would be better if it were looking at all three conditions.  Otherwise, this figure and the section of text describing it can just be referenced to the previously published article.

d.     Section 5 leaves out SYS entirely in the summary of cold-stress and the syndromes. Furthermore, the section itself focuses almost entirely on PWS in the concluding hypothesis paragraph. If this is truly a review on all three syndromes then the other syndromes should be included in summary sections as well.

2.     Lines 206-207: Anyhow management of PWS and SYS is mostly symptomatic and there is no cure as well as the mechanistic explanation of the disease is still elusive. This sentence has poor grammatical structure with two different thoughts connected. Use of anyhow to start a sentence is also not usually done in scientific writing. Other examples of poor English grammar are present in the manuscript, (even the next sentence following this one where “it” is used twice in the sentence making it difficult for the reader to know what “it” they are referring to), so an editorial review of the grammar with analysis for the clarity of each sentence should be done.

3.     Thermosensory abnormalities in Prader-Willi syndrome and autism spectrum disorder: is the Graal been underlooked? Sentence is not clear (lines 214-215)

a.     “Is” should be “Has”?

b.     Is the author referring to the Holy Grail?  No other reference to “Graal” is in the manuscript.

4.     The manuscript focuses more on PWS than the other two disorders. Table 1 gives a nice comparison of the three disorders, and this would seem a really go place to start a comparison/contrast of the three disorders and Oxt dysregulation (or not), perhaps then speculate on how Oxt dysregulation occurs in each or if missing in other, why it is not dysregulated in one versus the other.

5.     Lines 415-416: Another instance whether the text is misleading. “We concludes that TRPV1 mediates the pain 415 signaling of Oxt in neurons [77, 83, 84].”. First, grammatically, it should be “conclude”.  Second though, the format of the sentence suggest that Dr. Camerino’s laboratory did this work, but a look at references 77, 83, and 84 reveal the work is from other laboratories. Thus, the sentence is misleading in its format, but perhaps the authors meant for these other references to support her previous work (reference 13). In this way, the sentence should state that more clearly.

Comments on the Quality of English Language

There are numerous grammatical errors, and in some places the sentences are long and not clearly written.  

Author Response

                                                                                           December 28 2023

Reviewer # 2

This review article on oxytocin is part of a special issue on Oxytocin in IJMS. Dr. Camerino has nine other publications on oxytocin, the oxytocin knockout mice, and the oxytocin receptor, with four of these as review papers.  The paper is poorly written and  does not fully provide an overview of what is known about Oxt in Prader-Willi syndrome (PWS), Schaaf-Yang Syndrome (SYS) and Autism Spectrum Disorder (ASD), instead focusing more on PWS, and having significant overlap with another manuscript published by the same author in 2023. Suggestions and comments below are provided.

Dear respected reviewer number 2 we are glad you are following the progression of our studies from basic to translational. This is our second review regarding Prader-Willi syndrome and includes more data and references in comparison to the first one and also describe Oxytocin data new data in two other diseases as Schaaf-Yang syndrome and autism spectrum disorder.

I would like to point out that there may be significant overlap with Dr. Camerino’s article and another one she published in 2023.  In reviewing Dr. Camerino’s previous work in the area, I came across a very similar article, PMCID: PMC10297258. Even within the abstract, the line in the abstract “Prader–Willi Syndrome (PWS) is a genetic neurodevelopmental disorder that is caused by either the 13 deletion of the paternal allele of 15q11-q13, maternal uniparental disomy of chromosome 15 or defects in the chromosome 15 imprinting centre and is characterized by hyperphagia with significant 15 risk of obesity, low skeletal muscle tone and a variety of maladaptive behaviours and autistic spectrum disorder (ASD) for this reviewed manuscript is nearly identical to a line in her published article “…is characterized by cognitive impairment, hyperphagia and low metabolic rate with significant risk of obesity, as well as a variety of other maladaptive behaviours and autistic spectrum disorder (ASD).

Dear respected reviewer #2 the line you report above has been rephrased.

  1. In addition, Figure 1 from both articles are very similar, with just the addition of the infantile phase of PWS included in this article being reviewed. The text in Figure 1 in this article does not appear to be formatted correctly with extra spacing in the words within the figure (i.e. O xt and C irculating)

Figure 1 has been integrated with 2 new panels and captures describing Schaaf-Yang syndrome and autism spectrum disorder. The text of the figure has been corrected.

  1. Using Figure 1 in both articles, the author goes on to describe nearly the same information for PWS and oxt.

Figure 1 has been added of three new panels in comparison of our first review regarding the syndromes.This is why is more complete in comparison to the figure you mention.

  1. Since this is an article on all three syndromes, the figure would be better if it were looking at all three conditions.  Otherwise, this figure and the section of text describing it can just be referenced to the previously published article.

Thank you for the comments that we appreciated and the figure has been changed accordingly.

  1. Section 5 leaves out SYS entirely in the summary of cold-stress and the syndromes. Furthermore, the section itself focuses almost entirely on PWS in the concluding hypothesis paragraph. If this is truly a review on all three syndromes then the other syndromes should be included in summary sections as well.

Thank you for this comments. The reason why this articles focus more on PWS than on the other 2 syndromes is because no measurement of Oxt in SYS are available to date. However SYS individuals are hypotonic like PWS and present an higher incidence of ASD. Oxt abnormalities together with sensory abnormalities have been found in ASD individuals. A deeper  study of ASD is not the scope of this articles and will be treated in a new article. This is to say that in the current literature there are more data on Oxt  in PWS that the other 2 syndromes this is why we describe PWS better that the other 2. However PWS, SYS and ASD  shows many common features and Oxt dysfunctions. This comments has been added in section 5 at line 461.

  1. Lines 206-207: Anyhow management of PWS and SYS is mostly symptomatic and there is no cure as well as the mechanistic explanation of the disease is still elusive. This sentence has poor grammatical structure with two different thoughts connected. Use of anyhow to start a sentence is also not usually done in scientific writing. Other examples of poor English grammar are present in the manuscript, (even the next sentence following this one where “it” is used twice in the sentence making it difficult for the reader to know what “it” they are referring to), so an editorial review of the grammar with analysis for the clarity of each sentence should be done.

Thank you for this comments. Line 206-207 has been rephrased.

  1. Thermosensory abnormalities in Prader-Willi syndrome and autism spectrum disorder: is the Graal been underlooked? Sentence is not clear (lines 214-215)
  2. “Is” should be “Has”?

Respected reviewer #2 “is” has been changed in “has”

  1. Is the author referring to the Holy Grail?  No other reference to “Graal” is in the manuscript.

Respected reviewer #2, you are right we refer to the Holy Grail.  The sentence has been rephrased.

  1. The manuscript focuses more on PWS than the other two disorders. Table 1 gives a nice comparison of the three disorders, and this would seem a really go place to start a comparison/contrast of the three disorders and Oxt dysregulation (or not), perhaps then speculate on how Oxt dysregulation occurs in each or if missing in other, why it is not dysregulated in one versus the other.

Thank you for this comments. Figure 1 has been integrated with 2 new panels on the other 2 syndromes as well and include information from Table 1.

  1. Lines 415-416: Another instance whether the text is misleading. “We concludes that TRPV1 mediates the pain 415 signaling of Oxt in neurons [77, 83, 84].”. First, grammatically, it should be “conclude”.  Second though, the format of the sentence suggest that Dr. Camerino’s laboratory did this work, but a look at references 77, 83, and 84 reveal the work is from other laboratories. Thus, the sentence is misleading in its format, but perhaps the authors meant for these other references to support her previous work (reference 13). In this way, the sentence should state that more clearly.

Lines 415-416 has been rephrased.

Comments on the Quality of English Language. There are numerous grammatical errors, and in some places the sentences are long and not clearly written.  

Thank you for this comments. We corrected majority of sentences. If anything is still missing we will correct it during the proof if this is the case.

Round 2

Reviewer 2 Report

Comments and Suggestions for Authors

This review article on oxytocin is part of a special issue on Oxytocin in IJMS, and I reviewed the first version in December. The authors have responded to my and the other reviewer’s suggested updates, but some additional concerns remain and are listed below.

1.     Figure 1 has been updated and is improved in that Schaaf-Yang Syndrome and ASD are now represented. However, the figure is still of poor quality overall with abnormal spacing of headers and other text that is not consistent with figure preparation guidelines.

2.     In the text referring to this figure, there is inconsistent referral to the figure panels.  Where the authors say “Figure 1. Panel (A) In healthy individuals the Oxt system…” could instead simply say, As shown in Figure 1A, in healthy individuals… (and so on for the other parts of the paragraph).

3.     There are still grammatical and wording issues in the manuscript, which lead to a lack of confidence that the article was produced with care. A thorough editorial analysis is needed, but here are some issues that I have found.

a.     Line 28: 1956 in a short article were they highlight… (where they highlight..)

b.     Lines 65-66: “These articles were written [by] performing a literature search using PubMed, the word PWS, the drug name all through the year 2021.” In addition to missing the word “by”, it is unclear what point the authors are trying to make here. Are they saying that none of these articles report about oxytocin?

c.      Lines 68-69: “…established progressively very recently.” Unclear phrasing. Are the authors saying there is new information since 2021?

d.     Line 80-extra space before Figure 1

e.     Line 104: “however Autism spectrum disorder is more frequent compared to Prader-Willi syndrome.” (frequently)

f.      Line 105: Autism spectrum disorder individuals…”, would be better phrased, Individuals with autism spectrum disorder…In general, it is better to refer to the condition this way, rather than describing the individual as an ASD individual.

g.     Lines 107-108: “In Autism spectrum disorder Oxt appears deregulated but differently from Prader-Willi syndrome individuals these abnormalities are not caused by hypothalamic syndrome”. The sentence appears to be missing punctuation or words, or may be two sentences.

h.     Line 110: Who are “these” patients—all of them, or only one of the conditions?

i.       Line 112: Grow hormone (GH) -should be Growth hormone

j.       Line 160 “Last SYS individuals show a higher incidence of ASD compared to PWS”-do the authors mean “Lastly…”

k.     There are so many I cannot spend the time continuing to edit this.

4.     The authors continually state in the manuscript and table that there is no data on Schaaf-Yang patients and oxytonin. However, these statements are not entirely true, or at least the authors could do a better job of examining the mouse models, and patients case studies.  A search of PubMed with the words “Schaaf-Yang” and “oxytocin” reveals several interesting papers that could be used to support the authors view of the role of oxytocin in SYS, PWS, and ASD.  The mouse models of PWS are discussed thoroughly so it is not clear why the authors are not also considering what can be learned from the mouse models for SYS in terms of oxytocin.

a.     Ates et al (2019)-on the mouse model of Magel2 KO and oxytocin neurons outgrowth.

b.     Gigliucci et al., (2023) on mouse model rescue of Magel 2 KO by Oxt administration.

5.     Table 1 is not useful in its current form, which is essentially paragraphs of information in a tabular format. 

6.     In Table 2, how is the paraventricular nucleus different from the hypothalamus in your model?  Are you considering the whole hypothalamus versus a portion of it? Table 2 might also be better presented as a model of findings, than a table.

7.     The story on autism spectrum disorder and oxytocin is downplayed in the article, but there are over 650 papers on this, including articles detailing OXT and OXTR variants associated with ASD.  A newer report just available in January 2024 looks at peripheral Oxt levels and gray matter volume from individuals with ASD, which is very relevant to the article.

Comments on the Quality of English Language

Extensive English language editing needs to be done for clarity in nearly every paragraph of the manuscript. I have provided some edits, but gave up trying to help after line 160.

Author Response

This review article on oxytocin is part of a special issue on Oxytocin in IJMS, and I reviewed the first version in December. The authors have responded to my and the other reviewer’s suggested updates, but some additional concerns remain and are listed below.

Respected Reviewer # 2

First upfront explanation: We suggest this reviewer to  read carefully the title  “The pivotal role of Oxytocin’s mechanism of thermoregulation in Prader-Willi Syndrome, Schaaf-Yang Syndrome and Autism Spectrum Disorder”

As clearly  stated in the title this article do not regard the general role of oxytocin in autism nor in the rest of the syndromes but this articles discuss  exclusively the role of oxytocin regulation of thermogenesis in PWS, SYS and ASD. This article is not intended to include and reference  the rest of the 650 articles actually present in the pubmed on the topic “oxytocin and autism” that is a hugely wide topic and cannot be summarized in one article. This is clearly stated since my first submission in Dicember at line 278-279. Indeed the subject of oxytocin regulation of thermogenesis and related sensory deficits  is so novel that many readers  do not even consider thermogenesis to be a matter by itself. I hope that after this brief explanation will be  clear to you why I metaphorically called this findings  “The Graal”, or “Holy Graal” that according to me has been undelooked by the scientific community referring to and exclusively to oxytocin regulation of thermogenesis and NOT to other characteristics of PWS, SYS and ASD,  that are also important but not pertinent to this concept, at line 222-223. Also you raised the point that “the mouse models of PWS are discussed thoroughly so it is not clear why the authors are not also considering what can be learned from the mouse models for SYS in terms of oxytocin”. The answer to this question is again that the aim of this article is not to describe the animal models related to PWS, SYS and ASD unless they have a thermosensory deficit. Indeed the models described in the 2 papers you suggest Ates 2019 and Gilgiucci 2013 do not present a thermoregulation and sensory deficit. We added these references Ates et al (2019) and Gigliucci et al., (2023)  at line 82.

Second upfront explanation: Regarding the extensive English language editing that you provided giving up at line 160. Please note that trivial editing like space between words that for example you raise at line 80 Line 80-extra space before Figure 1” will be corrected during the proof and formatting by the journal.

Figure 1 has been updated and is improved in that Schaaf-Yang Syndrome and ASD are now represented. However, the figure is still of poor quality overall with abnormal spacing of headers and other text that is not consistent with figure preparation guidelines.

Reply: The figure 1 has been revised several times. So can you please highlight clearly where you think that is not consistent with figure preparation guidelines? Other minor editing like spacing between words  will be addressed during the proof.

In the text referring to this figure, there is inconsistent referral to the figure panels.  Where the authors say “Figure 1. Panel (A) In healthy individuals the Oxt system…” could instead simply say, As shown in Figure 1A, in healthy individuals… (and so on for the other parts of the paragraph).

Reply: This has been changed in the text following your advice.

There are still grammatical and wording issues in the manuscript, which lead to a lack of confidence that the article was produced with care. A thorough editorial analysis is needed, but here are some issues that I have found.

 Line 28: 1956 in a short article were they highlight… (where they highlight..)

Reply: Were has been changed in where

Lines 65-66: “These articles were written [by] performing a literature search using PubMed, the word PWS, the drug name all through the year 2021.” In addition to missing the word “by”, it is unclear what point the authors are trying to make here. Are they saying that none of these articles report about oxytocin?

Reply: By has been added. I’m not saying that none of these articles report about oxytocin but I’m saying that articles regarding thermoregulation and muscle contraction have been published after 2021. The sentence has been rephrased like this “These articles were written by performing a literature search using PubMed, the word PWS, the drug name including Oxt  all through the year 2021”.

Lines 68-69: “…established progressively very recently.” Unclear phrasing. Are the authors saying there is new information since 2021?

Reply: Yes, new information on thermoregulation and sensory deficits has been published after 2021 and the sentence has been rephrased like this “However, the new knowledge that Oxt regulates muscle mass, decreases proteolysis in skeletal muscle [24, 25] and triggers muscle contraction by thermoregulation [13, 26] established  very recently and after the year 2021”

Line 80-extra space before Figure 1

Reply: The extra space has been deleted. This minor editing will be finalize during the proof.

Line 104: “however Autism spectrum disorder is more frequent compared to Prader-Willi syndrome.” (frequently)

Reply: I cannot change frequent with frequently because it would change the sense of the sentence.  I change the sentence like this “Autism spectrum disorder is present more frequently in Schaaf-Yang syndrome compared to Prader-Willi syndrome.

Line 105: Autism spectrum disorder individuals…”, would be better phrased, Individuals with autism spectrum disorder…In general, it is better to refer to the condition this way, rather than describing the individual as an ASD individual.

Reply: Line 105 has been changed as you advise however I cannot change following this suggestion  all over the article because for example “IN INdividuals with ASD” as you advise to correct  sounds redundant because of the presence of double IN IN comparing to “in ASD individuals”.

Lines 107-108: “In Autism spectrum disorder Oxt appears deregulated but differently from Prader-Willi syndrome individuals these abnormalities are not caused by hypothalamic syndrome”. The sentence appears to be missing punctuation or words, or may be two sentences.

Reply: The sentence has been changed to this: However differently from Prader-Willi syndrome individuals these abnormalities in autism spectrum disorder are not caused by hypothalamic syndrome.

Line 110: Who are “these” patients—all of them, or only one of the conditions?

Reply: These has been corrected to PWS

Line 112: Grow hormone (GH) -should be Growth hormone

Reply: Corrected to Growth

Line 160 “Last SYS individuals show a higher incidence of ASD compared to PWS”-do the authors mean “Lastly…”

Reply: Corrected to Lastly

There are so many I cannot spend the time continuing to edit this.

Reply: The rest of the minor editing will be done during the proof.

The authors continually state in the manuscript and table that there is no data on Schaaf-Yang patients and oxytonin. However, these statements are not entirely true, or at least the authors could do a better job of examining the mouse models, and patients case studies.  A search of PubMed with the words “Schaaf-Yang” and “oxytocin” reveals several interesting papers that could be used to support the authors view of the role of oxytocin in SYS, PWS, and ASD.  The mouse models of PWS are discussed thoroughly so it is not clear why the authors are not also considering what can be learned from the mouse models for SYS in terms of oxytocin.

Ates et al (2019)-on the mouse model of Magel2 KO and oxytocin neurons outgrowth.

Gigliucci et al., (2023) on mouse model rescue of Magel 2 KO by Oxt administration.

Reply: The authors are not also considering what can be learned from the mouse models for SYS in terms of oxytocin because these mouse models do not have a thermosensory deficit. The aim of this study is not to describe animal models unless they have a thermosensory deficit as explained at line 227-228. From my personal literature search I learned that Oxt have never been measured in SYS  individuals. Do you know if Oxt has ever been measured in SYS?. We added thes references Ates et al (2019) and Gigliucci et al., (2023)  at line 82.

Table 1 is not useful in its current form, which is essentially paragraphs of information in a tabular format. 

Reply: Table 1 is not paragraphs in a tabular format because such paragraphs are explained at lines 74-162. Also reviewer #1 asked for additional tables this is why we integrated this new table.

I report what you wrote in your first revision regarding table 1:

  1. Table 1 gives a nice comparison of the three disorders, and this would seem a really go place to start a comparison/contrast of the three disorders and Oxt dysregulation (or not), perhaps then speculate on how Oxt dysregulation occurs in each or if missing in other, why it is not dysregulated in one versus the other.

The comparison/contrast on the 3 disorders that you advice in your first revision has been addressed at line 147-162. In this first revision you never mention that Table 1 is not useful in the current form rather you write that it gives a nice comparison of the 3 disorders and give a positive comment about it. I did not change Table 1 between your first revision and the second so why/where you contradict yourself and changed your opinion in this second revision regarding Table1?

  1. In Table 2, how is the paraventricular nucleus different from the hypothalamus in your model?  Are you considering the whole hypothalamus versus a portion of it? Table 2 might also be better presented as a model of findings, than a table.

Yes, the paraventricular nucleus is different from the hypothalamus in our model as previously shown by histomorphometry and described  in Ref 13..Table 2 has been presented as a  model findings in our paper referenced as  13 this is why we presented in the novel form of a  table in this review. This has been added in the text. “The cold stress model is explained in Table 2 and has been previously shown [13]” at line 330.

  1. The story on autism spectrum disorder and oxytocin is downplayed in the article, but there are over 650 papers on this, including articles detailing OXT and OXTR variants associated with ASD.  A newer report just available in January 2024 looks at peripheral Oxt levels and gray matter volume from individuals with ASD, which is very relevant to the article.

Reply: The scope of this article is not to discuss Oxt in ASD but is Oxt and thermoregulation and sensory deficits in ASD. Maybe in  a next article I’ll review the literature on ASD and Oxytocin but not in this one. The new reference you mention above has been added at line 87.

Extensive English language editing needs to be done for clarity in nearly every paragraph of the manuscript. I have provided some edits, but gave up trying to help after line 160.

Additional minor editing like space between words and spelling of words  will be addressed during the proof and formatted by the journal.

Round 3

Reviewer 2 Report

Comments and Suggestions for Authors

I appreciate your attention to the edits. 

I would still like to see an improved (i.e. re-done) version of Figure 1. The spacing and irregular font is not going to be fixed by the journal editorial staff. You need a clean copy as a jpeg or tiff that will be placed into a published version.  

Comments on the Quality of English Language

These have been fixed.

Author Response

Dear respected reviewer #3, Figure 1 has been revised upon your advices. We
thank you very much for the careful revision of our manuscript and we now
acknowledge the importance of adding reference 81 as you suggested.
However reference 81 has been published after our first submission in
December meaning that this a very hot topic in this field.

Reference 79, 80, 81 has been integrated correctly in the text.